# Single-cell Masked Autoencoder: An Accurate and Interpretable Automated Immunophenotyper

**Jaesik Kim,**[*] **Matei Ionita,**[*] **Matthew Lee, Michelle McKeague, Ajinkya Pattekar,**
**Mark Painter, Joost Wagenaar, Van Truong, Dylan T. Norton, Divij Mathew,**
**Yonghyun Nam, Sokratis Apostolidis, Cynthia Clendenin, Patryk Orzechowski,**
**Sang-Hyuk Jung, Jakob Woerner, Yidi Huang, Nuala J. Meyer,**
**Allison R. Greenplate,**[†] **Dokyoon Kim,**[†] **E. John Wherry**[†]

Institute for Immunology and Immune Health
University of Pennsylvania
Philadelphia, PA 19104
{jaesik.kim, matei.ionita,
Allie.Greenplate, Dokyoon.Kim, wherry}@pennmedicine.upenn.edu

## Abstract

High-throughput single-cell cytometry data are crucial for understanding the immune system's role in diseases and treatment response. However, the prevailing methods used for analyzing cytometry data, specifically manual gating and clustering methods, have certain limitations with scalability, robustness, and accuracy. In this study, we propose a single-cell masked autoencoder (scMAE), which offers an automated solution for immunophenotyping tasks such as cell type prediction. Our model aims to preserve the cell type definitions designed by the user, making interpretation and cross-study comparisons more accessible. The scMAE model follows a pre-train and fine-tune paradigm. During pre-training, scMAE utilizes Masked Single-cell Modelling (MScM) to learn relationships between protein markers in immune cells without the need for prior labeling information. Subsequently, the scMAE is fine-tuned on multiple specialized tasks, using a smaller designated portion of labeled data. Through evaluation experiments, we demonstrated that the pre-trained scMAE overcomes limitations of manual gating and clustering methods, providing accurate and interpretable cellular immunophenotyping. The introduction of scMAE represents a significant advancement in immunology research, enabling prediction and interpretation of cellular-level in immune disease.

## 1 Introduction

High-throughput, single-cell protein expression data as acquired through flow and mass cytometry are essential to understanding the immune system's role in infectious diseases, autoimmune diseases, or cancer, and its response after treatment. Cytometry assays typically profile millions of cells from a biological sample, allowing scientists to quantify cell-type specific biomarkers, even for rare cell types. For example, we can see which cell populations are differentially abundant, or which proteins are differentially expressed between subject groups. Immune profiling maps the similarity and diversity of the immune landscape for all subjects and patients, supporting individual-level prediction and precision medicine in the clinic.

Currently, the predominant approach for analyzing cytometry data is manual gating: the application of sequential filters to bivariate plots of protein markers to focus the analysis on particular cell

---

[*]These authors contributed equally
[†]Corresponding authors

NeurIPS 2023 AI for Science Workshop.

subsets of interest [14]. These bivariate plots visually represent the distribution of protein markers, allowing a human analyst to manually identify and select cells based on their prior knowledge of the marker distribution. However, manual gating has several serious drawbacks [15, 17]. First, it is time-consuming for panels larger than a dozen markers, as the number of 2D plots increases quadratically with the number of markers. Second, results from manual gating may be difficult to reproduce. Researchers' diverse gating strategies encompass distinct gating sequences, shapes, and boundaries which may affect robustness and replicability of cell subsets. The level of gating stringency also varies between individuals, contributing to inconsistencies in results. Third, manual gating workflows fail to use all information available in the data, because they only consider two dimensions at a time, rather than the full multivariate protein expression profile.

The ability to measure multiple protein markers simultaneously has led to high-dimensional data, prompting the development of automated analysis techniques, particularly unsupervised clustering methods like FlowSOM [21], PhenoGraph [12], Scaffold Maps [19], and X-shift [18]. While clustering approaches address the shortcomings of manual gating and offer speed, they also come with their own set of constraints. First, although unsupervised clustering methods can detect variability in the data, they cannot distinguish between biological or technical origin. Consequently, clustering methods are sensitive to batch effects, data distribution shifts and non-specific binding of antibodies [10]. Another challenge arises in cross-study comparisons, where slight changes in panel choice, dataset specifics, or stochastic elements can lead to discontinuous changes in cluster boundaries. For example, CD4 T cells might cluster differently based on memory subtype in one study and functional subtype in another, making a direct comparison challenging. To strike a balance between labor-intensive manual analysis and unpredictable unsupervised analysis, we focus on a combination of unsupervised and supervised learning to develop an automated method in this study. This has the advantage that our automated method can predict the immunophenotype of cells in future samples while using the same cell type ontology found in the training data.

In this study, we propose an accurate and interpretable automated immunophenotyper for single-cell cytometry data through *Masked Single-cell Modelling (MScM)*, which uses self-supervised pre-training techniques on single-cell cytometry data. During *MScM*, our model learns the relationships and dependencies between markers in immune cells by identifying expression patterns in the massive amount of data itself, without any additional labels. The pre-trained model can then be exported using a useful representation, giving it an advantage over using the original data in many downstream tasks. We show here that our model can overcome the limitations of manual gating and clustering methods. Our model **accurately** identifies complex cell types and offers **interpretability** on which protein markers it paid attention to when predicting targets. We will also demonstrate additional properties such as scalability and reproducibility. While several pre-trained models for single-cell RNA sequencing (scRNA-seq) data have been published in similar approachs [20, 8, 24, 7, 6], to our best knowledge, this is the first model pre-trained using single-cell cytometry data. To the best of our knowledge, our model is the first pre-trained model using single-cell cytometry data. This approach to immunophenotyping will have the potential to advance the field of immunology by extending it to predict and explain the cellular-level and individual-level phenotypes of various immune diseases.

## 2   Single-cell Masked Autoencoder (scMAE) Algorithm

We propose scMAE, a single-cell Masked Autoencoder model that constructs and employs latent embeddings of single-cell cytometry data to obtain state-of-the-art performance on several cell-level tasks. scMAE is built upon a Masked Autoencoder (MAE) [9] backbone structure, consisting of stacked transformer blocks in both encoder and decoder. Drawing from established practices in the domains of computer vision and natural language processing, scMAE is trained in two stages: self-supervised pre-training and supervised fine-tuning (Figure 1a). The main advantage of this approach is leveraging large-scale, easily obtainable unlabeled data in the first stage, while requiring smaller amounts of labor-intensive labeled data in the second stage. During pre-training, a random subset of the protein expression data is masked and fed to an encoder which produces latent embeddings of the masked data. In turn, these embeddings are fed to a decoder that attempts to reconstruct the unmasked, original data (Figure 1b, Figure P.1). The encoder-decoder system learns to optimize the embeddings to minimize reconstruction error. The true goal of pre-training is to obtain informative embeddings of the data, and the reconstruction of masked data allows the model to accomplish this goal even in the absence of any ground truth labels. During the second fine-tuning stage, the full

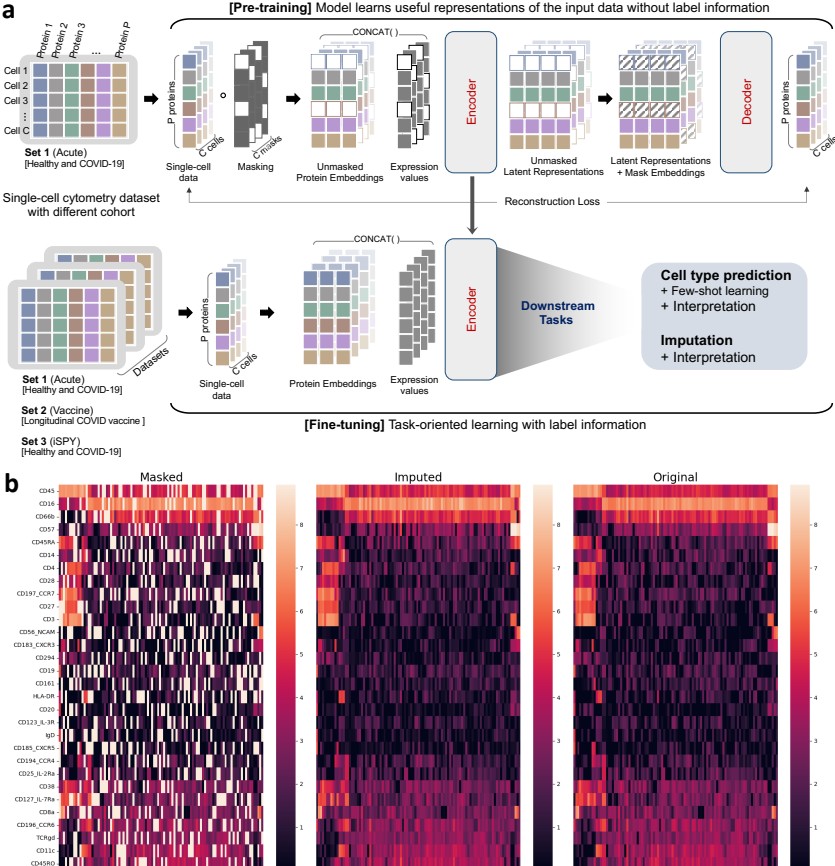

Figure 1: (a) Overview of Single-cell Masked Autoencoder (scMAE). In the pre-training step, protein expression data is randomly masked. Unmasked protein expressions are concatenated with its learnable protein embeddings and inputted into the encoder. The encoder generates unmasked latent representations, and they are combined with the learnable mask embeddings and fed to the decoder for reconstruction of the masked values. In the fine-tuning step, the pre-trained encoder produces latent representations for cells, facilitating cell-level downstream tasks. (b) From left to right, masked, imputed (reconstructed), and original data. Each row represents a marker protein, and each column represents a randomly sampled cell. The original data was 25% randomly masked, and those regions are colored white in the masked data. The masked regions are reconstructed accurately through scMAE.

protein expression data without masking is used to generate latent cell representations through the pre-trained encoder in the first stage. This can then be used for several downstream tasks, which may or may not require labeled data. Cell representations generated by the pre-trained encoder can be used for unsupervised tasks or plugged into another classifier to solve tasks through supervised fine-tuning. Specifically, we tested on two cell-level tasks using the pre-trained scMAE: cell type prediction, and imputation.

## 2.1 Masked Single-cell Modeling (MScM)

scMAE learns to maximize

$$P(V_{i,masked}|V_{i,unmasked}, \mathbf{E}_{unmasked}) \tag{1}$$

where $i$ indexes cells, $V_{i,masked} \in \mathbb{R}^{r \cdot p \times 1}$ denotes masked protein expressions of cell $i$, and $\mathbf{E}_{masked} \in \mathbb{R}^{r \cdot p \times (d-1)}$ denotes masked protein embeddings after masking. $r$ is a masking ratio, $p$ is the number of proteins in the data, and $d$ is a hidden dimension size. Likewise, $V_{i,unmasked} \in$

$\mathbb{R}^{(1-r)\cdot p\times 1}$ denotes unmasked protein expressions of cell $i$, and $\mathbf{E}_{unmasked} \in \mathbb{R}^{(1-r)\cdot p\times(d-1)}$ denotes unmasked protein embeddings.

The encoder ($f_e$) generates a latent representation of the cell. The unmasked latent representation $\mathbf{H}_{i,unmasked} \in \mathbb{R}^{(1-r)\cdot p\times d}$ of cell $i$ is defined as the following,

$$\mathbf{H}_{i,unmasked} = f_e((\mathbf{E}_{unmasked} \parallel V_{i,unmasked}) + \mathbf{P}_{unmasked}), \tag{2}$$

where $\mathbf{P}_{unmasked} \in \mathbb{R}^{(1-r)\cdot p\times d}$ is sine-cosine positional embeddings for masked proteins. The idea of the concatenation ($\parallel$) of protein embeddings with expression values was inspired from MET [16].

The decoder ($f_d$) reconstructs the masked values as following,

$$\hat{V}_{i,masked} = f_d((\mathbf{H}_{i,unmasked} \parallel \mathbf{M}) + \mathbf{P}) \tag{3}$$

Let $M$ denote a learnable mask token embedding represented as a row vector $M \in \mathbb{R}^{1\times d}$. We construct a matrix $\mathbf{M}$ by stacking this vector $rp$ times, such that the resulting matrix $\mathbf{M}$ has dimensions $(r \cdot p \times d)$. $\mathbf{P} \in \mathbb{R}^{p\times d}$ are sine-cosine positional embeddings. To calculate the reconstruction loss, we use mean square error (MSE) loss for all cells,

$$Loss = \sum_i MSE(\hat{V}_{i,masked}, V_{i,masked}) \tag{4}$$

## 2.2 Datasets

We analyzed Cytometry by time of flight (CyTOF) data originating from three distinct COVID-19 studies conducted at the University of Pennsylvania. These data were acquired with a 30-marker panel and are referred to as the Acute dataset, Vaccine dataset, and iSPY dataset. The Acute dataset includes 6.5M cells from the 26 individual single-cell cytometry files, corresponding to 13 COVID-19 patients and 13 healthy individuals. The Vaccine dataset was obtained longitudinally from individuals before and after vaccination for SARS-CoV-2. This dataset is composed of 36.7M cells across 150 files, measuring cells from 44 individuals at 4 timepoints. Lastly, the iSPY dataset was obtained longitudinally from 42 COVID-19 infected individuals at the time of admission and after one week of treatment. It includes 11.9M cells from 56 measurements. We used the Acute dataset for pre-training and used all the three datasets in the downstream evaluations. Each of the datasets underwent a standard manual cleanup procedure, which involves removing aggregates, debris, doublets, beads, and dead cells from the data (see Appendix B).

## 3 scMAE is an Accurate Cell Immunophenotyper

Cell type annotation is the primary outcome of manual gating and clustering methods. Our model can perform automated cell type prediction on single cell datasets by fine-tuning the model with cell type labels. A total of 46 cell types obtained for each cell from manual gating were used as ground truth labels (Figure P.2). We used 60% of the Vaccine data for training, 20% as validation and the remaining 20% as an internal test set. We used the iSPY dataset and the Acute dataset as external sets (External set 1 and 2, respectively). We compared scMAE with a gradient boosting decision tree (GBDT) [4], a fully connected deep neural network (DNN), and a convolutional neural network (CNN) (see Appendix E) as well as cytometry-specific analysis methods: static gating and unsupervised clustering with FlowSOM.

Static gating is a baseline method which involves applying the filters manually defined in the training data to the testing datasets, without adjustments for inter-sample variability (Figure P.3). This is equivalent to manually constructing a decision tree and then applying it on the testing data. The other supervised models used here can be seen as refinements of this idea: they attempt to learn a more robust encoding of the gating information, by using multivariate rather than bivariate expression patterns. In the case of scMAE, it uses masking in the pre-training stage. Alongside the supervised classification methods, we include FlowSOM, a popular unsupervised clustering method for cytometry. To match our supervised paradigm, we add an inference mode to FlowSOM by mapping each unseen test datapoint to the nearest SOM node (see Appendix E).

Since most of the cells, over 60%, are Neutrophils, an imbalanced cell type distribution, rather than using Accuracy as an evaluation metric, we used Balanced accuracy (Bacc) to give a fair assessment

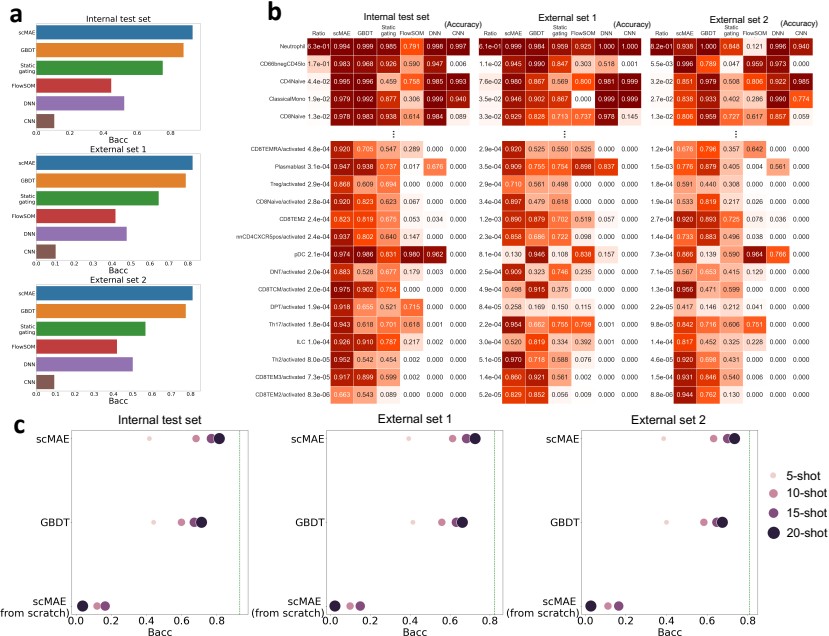

Figure 2: (a) Model comparisons in the cell type prediction. (b) Accuracy of cell type predictions for 5 abundant and 15 rare cell types. Deep neural nets (DNN) denotes the fully-connected neural network proposed by Cheng, L. *et al.* [5] and Li, H. *et al.* [13] for cytometry data analysis. Convolution neural nets (CNN) denotes a model architecture that removes only pooling layer from the CNN proposed by Hu. Z *et al.* [10] for cytomegalovirus (CMV) classification. (c) Few-shot learning performance for cell type prediction. Each green dashed line represents the performance of the full fine-tuned scMAE when used all available training set, reported in the (a).

of imbalanced label. The experimental results showed consistently high Bacc on both internal test sets and two external sets (Figure 2a). The internal test set showed a 93.1% Bacc, while the external sets showed 82.5% and 81.0% Bacc, respectively. When we looked at performance by cell type, we found that our model is more accurate than others for most cell types (Figure P.4). In addition, the fine-tuned scMAE from the pre-trained outperformed the scMAE from scratch (non-pre-trained), which demonstrates the benefit of pre-training (Table O.1).

Our model performed particularly well on rare cell types. Accurate prediction of rare cell types is difficult because it is easy for a model to be trained with a bias toward more frequent cell types. However, when comparing performance on cells with a frequency of less than 0.1% in Figure 2b, both internal test set and external sets show more accurate predictions for rare cell types than the comparison models in most cases.

FlowSOM scores lower on our accuracy metrics, as expected, because it makes no use of the training labels. We included it in the comparison to illustrate one important pitfall of unsupervised analysis: it uncovers true variability in the data, which nonetheless may not be biologically interesting (for example, splitting the dominant population of neutrophils into 6 clusters, based on non-specific binding of anti-CD3 or anti-TCRgd), while missing subtle but biologically meaningful differences (for example, differences between EM1, EM2, EM3 T cell phenotypes, based on CD27 and CCR7 expression).

These results show that our model is robust to technical variation between datasets, even when applied to the analysis of datasets derived and processed separately. For example, the cell immunophenotyping model was trained on the Vaccine dataset, run on frozen samples from healthy subjects in 2021, and it performed better than all other methods on the Acute dataset, run on fresh samples from subjects with acute COVID in 2020.

# 4    scMAE is a Few-shot Learner

Unlike full fine-tuning, few-shot learning trains a model with a limited amount of training data. $N$-shot uses only $N$ samples for each class in the classification problem. A pre-trained large language model trained by self-supervised learning is known as a good few-shot learner [3]. Similarly, our model was tested on a cell type prediction task for a few-shot learning setting. We ran 5-shot, 10-shot, 15-shot and 20-shot experiments. Training, validation, testing, and external testing sets are the same as in the previous cell type prediction tasks.

As expected, the pretrained scMAE approaches the performance of training with the full training set as the number of N-shots increases (Figure 2c). On the other hand, since the scMAE from scratch (non-pre-trained) has many parameters and no pre-trained information, we can see that it does not learn at all with small samples. It is worth noting that GBDT also performed reasonably well, but scMAE outperformed it based on the pre-trained knowledge. This shows that our pre-trained model can learn in the right direction even when there is very little labeled data for the new downstream task.

# 5    scMAE Enhances Regression Imputation

Current technology for flow and mass cytometry only allows a few dozen markers, and sometimes cost considerations may reduce the number even further. Traditionally, cytometry technologies were used by immunologists to answer very specific experimental questions which explains why a limited panel of markers and manual gating were sufficient at the time. However, it would be helpful to exploit high-dimensional patterns of protein expression to predict measurements from large panel sizes using only a smaller, cheaper panel. To investigate the feasibility of this, Becht, E. *et al.* [2] proposed Infinity Flow, applying a Gradient Boosting tree model [4] to impute the expression of over 300 markers using only 15. To test whether our cell latent representations can enhance regression imputation, we conducted experiments in which we masked 7 markers, including those commonly associated with memory subsets in T cells (CD27, CD28, CD45RA, CD45RO, CD127, CD197). Then we used the remaining information to predict the masked marker expressions using Infinity Flow and the scMAE with imputation supervised finetuning. We used the Acute data for training, and the Vaccine dataset and iSPY dataset as external sets (External set 1 and 2, respectively). R-squared is used as an evaluation metric.

The scMAE achieved moderate imputation performance (0.2-0.6 R-squared, Figure 3a), despite only having access to 23 markers not known to be predictive of T cell memory states and their associated masked markers. Overall, our method showed improvement over Infinity Flow for five of the seven markers. Moreover, scMAE learned more than just patterns of constitutively expressed proteins, such as CD45RA in NK cells and CD45RO in neutrophils. Correlations between true and predicted values are high even when restricted to T cells, or to CD27 expression in B cells (Figure 3b, Figure P.5,P.6,P.7). This suggests that our method infers information about T cell and B cell memory states, even in the absence of the standard memory markers.

# 6    scMAE is an Interpretable Immunophenotyper

A multi-head self-attention of the transformer blocks in scMAE enables interpretable predictions for downstream tasks. Attention scores show which marker information and relationships are important to the prediction tasks. High attention score means that marker information is used a lot from the other markers. We first measured the attention score of each feature for each cell type in cell type prediction. (Figure 4a) Notably, the marker with the highest attention score in all cell types is CD45, which distinguishes between the two main immune cell lineages: granulocytes and mononuclear cells. Aside from this, most markers were highly attended in cell types in which they are highly expressed: for example, CD19 in B cells, CD123 in basophils and pDCs, CD294 in basophils and eosinophils.

Similarly, we measured the attention score of 23 markers for each cell type used to predict the expression of 7 markers in the Imputation task (Figure 4b). For cell types where the masked markers are either constitutively expressed or constitutively not expressed, the model mostly attended to the available markers which determine the cell type (CD294, CD66b, CD45 for eosinophils; CD16, CD45 and, interestingly, HLA-DR for neutrophils). In the case of T cells, where knowing the cell

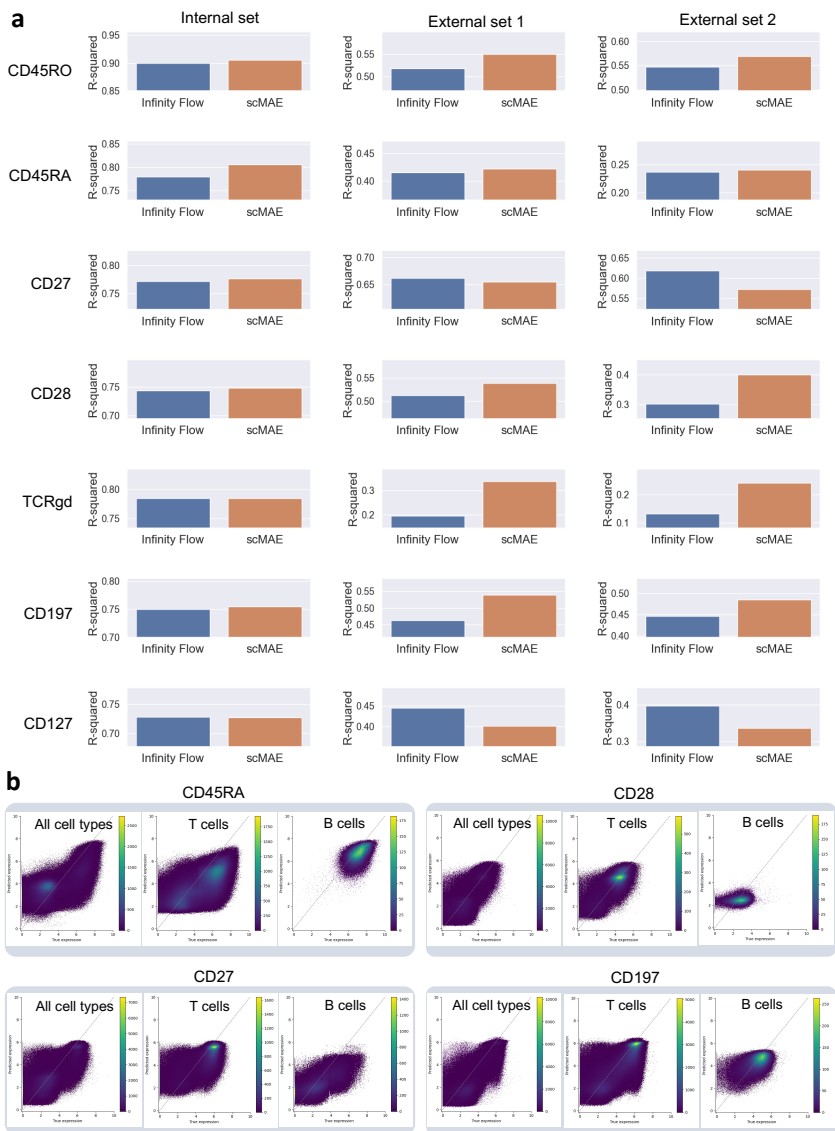

Figure 3: (a) R-squared comparison between Infinity Flow and scMAE in the imputation task. Total 7 markers were masked and predicted by the two models. (b) Plots of true expression and predicted expression for each marker in the external set (Vaccine dataset). The dashed line represents the ideal relationship as a reference line to assess the performance.

type is insufficient for predicting expression of the masked proteins, the model attended to the T cell marker protein CD3, but also to CD45 and HLA-DR, both of which are shown to be negatively correlated to CD45RA (Figure P.11).

This cell type-specific attention score shows a consistent pattern with external datasets (Figure P.8). In addition, the variance of attention scores between samples is not significant (Figure P.9,P.10). Overall, scMAE attention scores help both to confirm that known marker proteins were used by the model to make predictions, and to discover possibly unexpected correlations, such as those between CD45, HLA-DR and CD45RA.

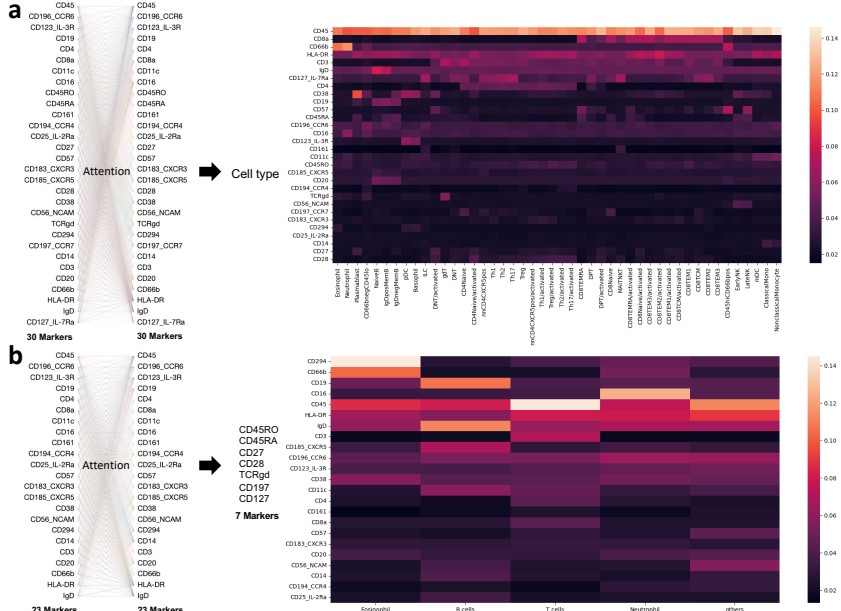

Figure 4: (a) Interpretation in the cell type prediction by the attention scores for the iSPY dataset (external set 1). The heatmap shows protein markers with high attention score as bright red for each cell type. (b) Interpretation in the imputation task by the attention score for the Vaccine dataset (external set 1). From 23 markers to impute the other 7 markers, it measures which input features have high attention from the other features during prediction. The heatmap shows the protein markers with high attention score as bright red for each cell type. For the left figure in (a) and (b), we used Bertvis [22] for visualization of attention weights.

# 7 Conclusion

In this manuscript, we introduced scMAE, a masked autoencoder model which builds latent embeddings of single-cell cytometry data and uses them to achieve good performance across a range of cell-level tasks. scMAE employs a training and inference paradigm that enhances scalability and reproducibility, outperforming alternative methods in making inferences on new datasets. Pre-training scMAE models with limited label information leads to improved performance, faster convergence, and stability, with potential for even greater gains by pre-training on larger and more diverse datasets. Especially, the fine-tuned scMAE is as accurate as manual gating, with the labor-free advantages of automated analysis. To the best of our knowledge, scMAE is the first such model which specializes on cytometry data. Our results are a proof of concept for applying a combination of unsupervised and supervised analysis in the training-inference paradigm to multiple cytometry datasets that use the same panel. The promise of this approach is that it generalizes easily to thousands of samples across multiple studies, providing robust and interpretable results while minimizing manual analysis.

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

## A  Single-cell cytometry and single-cell transcriptomics for immunology research

Single-cell cytometry, encompassing flow cytometry and mass cytometry (CyTOF), focuses on measuring the expression of a few dozen proteins at the individual cell level by tagging cells with fluorescent markers or metal isotopes, respectively. The data garnered primarily revolve around protein expression dynamics, cell phenotype distribution, or cell signaling pathways.

On the other hand, single-cell transcriptomics, employing techniques like single-cell RNA sequencing (scRNA-seq), delves into gene expression by sequencing the RNA of individual cells. The data here concern gene expression levels across thousands of genes per cell, paving the way for more complex modeling that can unveil deeper insights into gene expression patterns, regulatory networks, and the identification of new cell types or states.

However, the high cost, dimensionality, and computational demands associated with single-cell transcriptomics make cytometry a more practical choice for many immunology studies. Moreover, cytometry assays typically measure more cells than scRNA-seq by 1-2 orders of magnitude, offering better resolution into rare cell types, such as activated or antigen-specific lymphocytes.

## B  Preprocessing

Fresh (Acute dataset) or frozen (Vaccine and iSPY datasets) whole blood samples were stained with the MaxPar Direct Immune Profiling Assay and run on a CyTOF 2 instrument. After data acquisition, .fcs files were gated for beads, debris, doublets, and dead cells using the OMIQ platform; representative gates are shown in Figure P.12. After gating, DNA intercalator, viability, Gaussian and bead channels were dropped, and the remaining protein expression channels were transformed using inverse hyperbolic sine with a cofactor of 5.

In the Vaccine dataset, the definition of secondary immune response was defined as follows. We labeled a secondary immune response as "Yes" if it occurred after a healthy person received two vaccines, or after a person with COVID-19 received one vaccine, or after a person with COVID-19 received two vaccines. If a healthy person received a single vaccine, we labeled it "No".

## C  Model details

Transformer block

The transformer block consists of alternating layers of multihead self-attention (MSA) and multilayer perceptron (MLP) blocks (Equation 5). Layer norm (LN) [1] is applied before every block, and Drop path (DP) [11] is applied after every block. The MLP contains two linear layers with GELU activation function.

$$\mathbf{E}_l = \mathbf{E}_{l-1} + DP(MSA(LN(\mathbf{E}_{l-1}))) \quad (l = 1, \cdots, L), \tag{5}$$

$$\mathbf{E}_l = \mathbf{E}_l + DP(MLP(LN(\mathbf{E}_l))) \quad (l = 1, \cdots, L), \tag{6}$$

where $\mathbf{E}_{l-1}$ denotes output embeddings of the $(l-1)$-th layer and input embeddings of the $l$-th layer at the same time.

Multihead self-attention

In the multihead self-attention (MSA) layer, we compute query, key, and value matrix $(\mathbf{Q}, \mathbf{K}, \mathbf{V})$ from the input embeddings $(\mathbf{E})$ for each head (Equation 7) and compute $h$ heads by weighted sum of all values by attention weight for each head, where attention weight is calculated by the pairwise similarity between two elements of the input and their respective query and key representations (Equation 8). Finally, h heads are concatenated, and the output is linearly projected (Equation 9)

$$[\mathbf{Q}, \mathbf{K}, \mathbf{V}] = \mathbf{E}\mathbf{W}_{qkv}(\mathbf{Q}, \mathbf{K}, \mathbf{V} \in \mathbb{R}^{p \times d_h}), \tag{7}$$

where $\mathbf{E} \in \mathbb{R}^{p \times d}$ is input embeddings $\mathbf{W}_{qkv} \in \mathbb{R}^{d \times 3d_h}$ is learnable weight matrix, and $d_h$ is set to $d/h$.

$$Attention(\mathbf{Q}, \mathbf{K}, \mathbf{V}) = softmax((\mathbf{Q}\mathbf{K}^T)/\sqrt{d_h})\mathbf{V}, \tag{8}$$

$$MSA(\mathbf{Q}, \mathbf{K}, \mathbf{V}) = Concat(head_1, \cdots, head_h)\mathbf{W^O}, \tag{9}$$

where $head_i = Attention(Q_i, K_i, V_i)(i = 1, \cdots, h)$, and $\mathbf{W^O} \in \mathbb{R}^{d \times d}$ is linear weight matrix.

Single-cell Masked Autoencoder

The whole structure consists of an encoder and a decoder, which are used in the pretraining step. The encoder is only then used with a single linear layer in the downstream supervised finetuning. The encoder ($f_e$) consists of 12 layers of transformer blocks. Each block has 12 heads and 768 hidden dimensions. The number of parameters in the encoder is 85 million. The decoder ($f_d$) is smaller than the encoder. It consists of 4 layers of transformer blocks with 6 heads and 384 hidden dimensions for a total of 7 million parameters. The dimension size of latent cell representations for the downstream tasks is 768. This setting was proposed in the Masked autoencoder.

Why positional embedding is necessary

It might seem that positional embedding is not necessary because the input is a tabular data. However, the position serves as an index to indicate which protein's expression value should be reconstructed by the decoder during *MScM*. For example, if 2nd, 3rd, and 7th proteins of 10 proteins are masked, positional embedding provides information to reconstruct the expression of the 2nd, 3rd, 7th proteins. Therefore, when using scMAE, users make sure to match the order of the proteins.

## D  Training details

Cell representation

After pretraining, the cell representation ($C_i$) of cell $i$ is obtained as follows.

$$\mathbf{H}_i = f_e((\mathbf{E} \parallel V) + \mathbf{P}) \tag{10}$$

$$\mathbf{C}_i = \sum_k \mathbf{H}_i[k, :] \tag{11}$$

Then, this cell representation is used as input of a linear layer for cell-level downstream tasks.

### Supervised learning in downstream tasks

Cross entropy loss is employed for classification tasks and Mean squared error (MSE) loss is employed for regression tasks.

In the cell type prediction task,

$$\hat{y}_i = Linear(\mathbf{C}_i), \tag{12}$$

$$Loss_{CE} = -\sum_i y_i log\hat{y}_i, \tag{13}$$

where $y_i$ and $\hat{y}_i$ indicate the ground truth cell type and the predicted probability for cell type of cell $i$, respectively.

In the imputation task,

$$\hat{y}_j = Linear(\mathbf{C}_{j,unmasked}), \tag{14}$$

$$Loss_{MSE} = \sum_j (\hat{y}_j - y_j)^2, \tag{15}$$

where $y_j$ and $\hat{y}_j$ denote the ground truth expression value and the predicted value of masked protein $j$, respectively.

### Impact of Masking ratio during pre-training

To test if masking ratio affects scMAE training, we trained three different versions of the model with masking ratios of 0.25, 0.5, and 0.75. The result was there was no significant difference in performance on the cell type prediction tasks (Table O.2). Therefore, all the scMAE experiments were performed with a 0.25 masking ratio.

### Training setting

The configuration includes a batch size of 768, drop path regularization of 0.1, AdamW optimizer with momentum of 0.9 and weight decay of 0.05, learning rate of 0.0005 with a cosine scheduler, and label smoothing during fine-tuning.

### Computational cost in training and inference

The pre-training required 10 days with four of GeForce RTX 2080 Ti Rev. A to process 6.5M cells through 200 epochs. Fine-tuning the model for cell type prediction took 13 days on a single GeForce RTX 2080 Ti Rev. A GPUs to process 29.4 million cells through 100 epochs, with early stopping implemented. For inference, the runtime was 1.2 hours for 7.3M cells under the Vaccine dataset and 2.1 hours for 18.4M cells under the Acute dataset and iSPY dataset, both on a single GPU.

## E   Benchmarking models

### Manual gating

Each sample from all datasets was manually gated using the OMIQ platform to obtain the 46 terminal populations used as ground truth labels. A summary of the gating strategy is shown in Figure P.2.

### Static gating

For each gate in our hierarchy, we aggregated the candidate gate positions from all training samples in the Vaccine dataset into one consensus gate. By definition, a point is in the consensus gate if it falls into at least 30% of all the candidate gates (Figure P.3). We then created a consensus hierarchy out of all consensus gates and applied it statically to all test samples.

### FlowSOM clustering

An unsupervised FlowSOM clustering model was trained using the same 60% of the Vaccine data samples that were used for fine-tuning scMAE. Version 2.6.0 of the FlowSOM R package was used with default parameters, except for the total number of metaclusters, which we set to 46 to match the number of ground truth labels. As an unsupervised clustering algorithm, FlowSOM does not have an inference mode. We performed inference on testing datasets (20% of the Vaccine dataset as an internal test set, and the two external test sets) by assigning each datapoint to the nearest SOM node from the trained model, and preserving the assignment of nodes to metaclusters from the training

phase. Evaluation of accuracy and balanced accuracy required the extra information of a bipartite matching between the 46 FlowSOM clusters and the 46 ground truth labels. Following Weber, L. M. *et al.*[23], we obtained the matching using the Hungarian algorithm, implemented in the function *solve_LSAP* of the R package clue.

### Gradient Boosting Decision Tree (GBDT)

We used XGBoost [4] python package for GBDT. We ran XGBoost's regressor or classifier with 100 estimators and 0.03 learning rate and set early stopping based on the performance change for the validation set.

### Fully connected Deep Neural Network (DNN)

Cheng, L. *et al.* [5] and Li, H *et al.* [13] proposed a fully-connected neural network for cytometry data. Both were designed for a cell representation and cell-level prediction tasks, so we use this architecture as a comparison model.

### Convolutional Neural Network (CNN)

Hu. Z *et al.* [10] proposed a model using convolutional neural network for cytomegalovirus (CMV) classification. The original model was designed for subject-level tasks, but since it uses a CNN structure to draw cell representations and pool them, we modified to the same architecture without the pooling layer as a comparison model.

## F    Metrics

### Balanced accuracy (Bacc)

For a multi-class imbalanced dataset, we used Balanced accuracy (Bacc) instead of Accuracy. Balanced accuracy is defined as a macro-average of recall scores per class in a multi-class classification.

A Recall score is defined as:

$$Recall = TP/(TP + FN), \tag{16}$$

where $TP$ is true positive, and $FN$ is false negative.

### R-squared

In a regression task, if $\hat{y}_i$ is the predicted value of the $i$-th sample and $y_i$ is the corresponding true value for total $n$ samples, the R-squared is defined as:

$$R^2 = 1 - \frac{\sum_{i=1}^{n} (y_i - \hat{y}_i)^2}{\sum_{i=1}^{n} (y_i - \bar{y})^2}, \tag{17}$$

where $\bar{y} = \frac{1}{n} \sum_{i=1}^{n} y_i$.

### AUROC

A receiver operating characteristic (ROC) curve is widely used for evaluating prediction models. It plots True Positive Rate (TPR) against False Positive Rate (FPR).

$$TPR = \frac{TP}{TP + FN}, \tag{18}$$

$$FPR = \frac{TP}{FP + TN}, \tag{19}$$

Where TP, FP, TN, and FN are the number of true positives, false positives, true negatives, and false negatives respectively. AUROC stands for the area under the ROC curve.

## G    Few-shot learning setting in the cell type prediction

For $N$-shots, we trained using only the first $N$ samples per class in the training and validation sets and then evaluated on the entire test set. We compared performance for 5, 10, 15, and 20 shots.

## H  Imputation

We masked CD45RO, CD45RA, CD27, CD28, TCRgd, CD197, and CD127 expressions and used the remaining markers to predict the expression of these seven marker expressions. Infinity Flow used GBDT as the imputer. Similarly, the unmasked cell representations were first extracted from the pre-trained scMAE and used as input to GBDT to train and then evaluated on the external test sets (not end-to-end).

## I  Attention score

From Equation 8, we first obtain the output of $softmax$ function for the interpretation for cell $i$ (Equation 20) and calculate the attention score $W_i$ by averaging over the query axis (Equation 21).

$$\mathbf{A_i} = softmax \frac{\mathbf{Q_i K_i}^T}{\sqrt{d_h}} \tag{20}$$

$$W_i = \frac{1}{p} \sum_{k=1}^{p} \mathbf{A_i}[k,:] \tag{21}$$

In our experiments, we sampled 2% of all cells for each dataset. To calculate the attention score, we only used the information from the first layer, because the first layer is the most influential in determining which inputs to give attention to, and there was no significant difference in attention between inputs after the second layer.

## J  scMAE is a Scalable Learner

Due to the popularity and high throughput of cytometry assays, there is an abundance of high-dimensional cytometry data compared to other single cell modalities. The adoption of standardized panels, such as the one used in our three datasets, has led to a large number of datasets from many institutions that are directly comparable. Although manual gating remains the preferred classification approach among immunologists, this approach is too time- and labor-intensive to support the data analysis needs of multi-cohort and/or multi-institutional studies, as the scale can be in the over hundreds of millions or even billions. Our approach, on the other hand, can train on large-scale data since it is natively trained using a mini-batch approach. The time complexity in the training phase is linear in the number of samples. Also, scMAE can quickly and accurately make inferences on new datasets once the model has been trained.

## K  scMAE is a Reproducible Immunophenotyper

To enable a direct comparison of methods, we adopted a paradigm of training models and then using them to make inferences on a new dataset. This is not how manual gating or clustering methods are typically used: manual gating usually imports historical gates, which are then manually adjusted when necessary for each sample while clustering is used for discovering sources of variability in each dataset independently. The main advantages of the train-inference paradigm are scalability and reproducibility: any investigator can apply the exact same model to any dataset, obtaining results that are directly comparable. Our results show that scMAE outperforms alternative models within this paradigm.

## L  Advantages from the Pre-training

Comparing learning without pre-training (from scratch) and with pre-training, we found that first, learning without any label information was surprisingly effective in distinguishing between batch information and biological variation and clustering in agreement with cell type. Second, in task-oriented learning with specific labels, we could see not only performance improvement but also faster convergence in learning. Finally, even in the few-shot setting, our model was able to learn stably, while the from-scratch model was not able to learn at all with little training data.

In this study, we only pre-trained with one of the three available cohorts in order to evaluate the performance on several downstream tasks using the other two cohorts. In future work, we can pre-train on a much larger quantity of data from multiple studies, improving the power and robustness of the model. We hypothesize that it will be better able to distinguish between batch information and biological variation, and it will have a deeper understanding of protein functions and protein expression patterns. This, in turn, will lead to more robust and accurate predictions in various downstream tasks.

## M   Applicability of scMAE for flow cytometry data

We open-source the model parameters of the pretrained scMAE, as well as the fine-tuned scMAE for cell type prediction. While these models are trained on only CyTOF data, its application to flow cytometry data might not be recommended due to inherent technical differences. Specifically, the methodologies used in flow and mass cytometry yield disparate patterns of protein expression. However, if scMAE undergo pretraining specifically with flow cytometry data from scratch, it would indeed become a viable approach for flow cytometry datasets.

## N   Software

This project would not have been possible without numerous open-source Python packages including *torch, torchvision, timm, deepspeed, einops, jupyter, matplotlib, numpy, pandas, scikit-learn, seaborn, FlowCytometryTools, scipy,* etc. Specific versions for each package can be found at https://github.com/JaesikKim/scMAE/blob/master/requirements.txt.

## O   Supplemental tables

Table O.1: Balanced accuracy comparison between the non-pre-trained and the pre-trained scMAE in cell type prediction.

| Models | Internal test set (Bacc) | External set 1 (Bacc) | External set2 (Bacc) |
|---|---|---|---|
| scMAE from scratch | 0.925 | 0.822 | 0.810 |
| scMAE with fine-tuning | 0.931 | 0.825 | 0.810 |

Table O.2: Cell type prediction using different masking ratios in the scMAE.

| Masking ratio | Internal test set (Bacc) | External set 1 (Bacc) | External set2 (Bacc) |
|---|---|---|---|
| 0.25 | 0.931 | 0.825 | 0.810 |
| 0.5 | 0.931 | 0.824 | 0.813 |
| 0.75 | 0.932 | 0.825 | 0.812 |

## P   Supplemental Figures

Use the output of the masked protein's position to predict the masked values

Randomly mask 25% of proteins

Single-cell data input

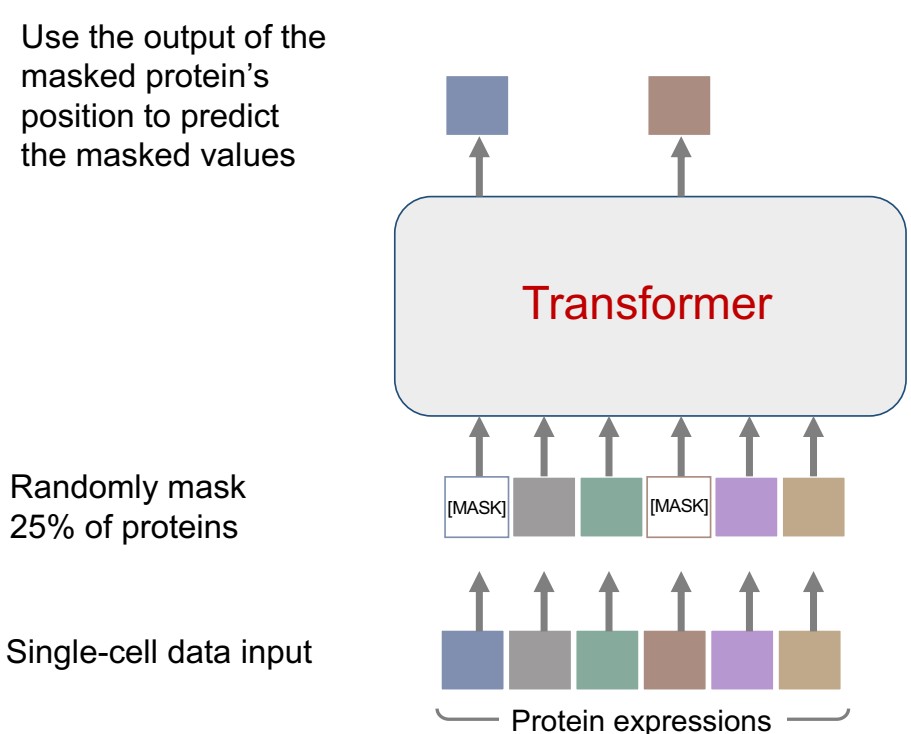

Figure P.1: Overview of Masked Single-cell Modelling (MScM)

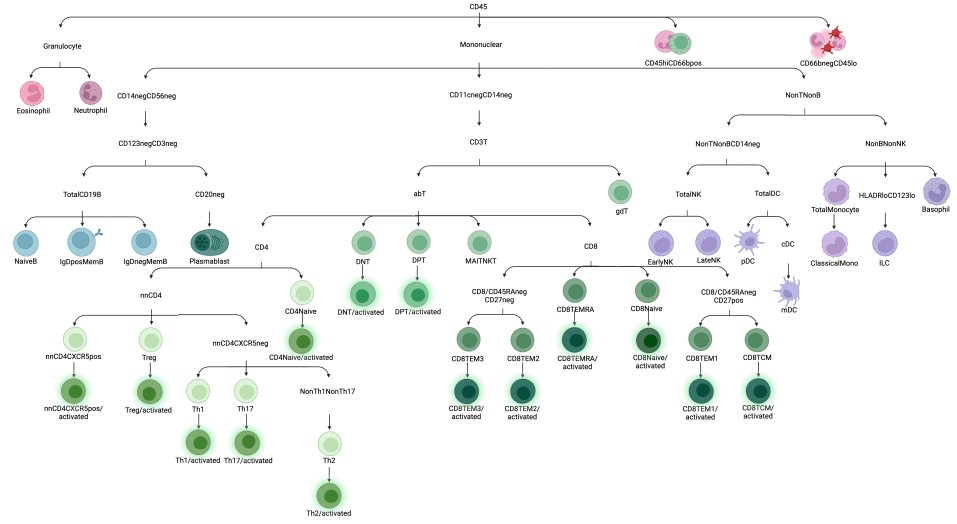

Figure P.2: Standard gating strategies for 46 cell types.

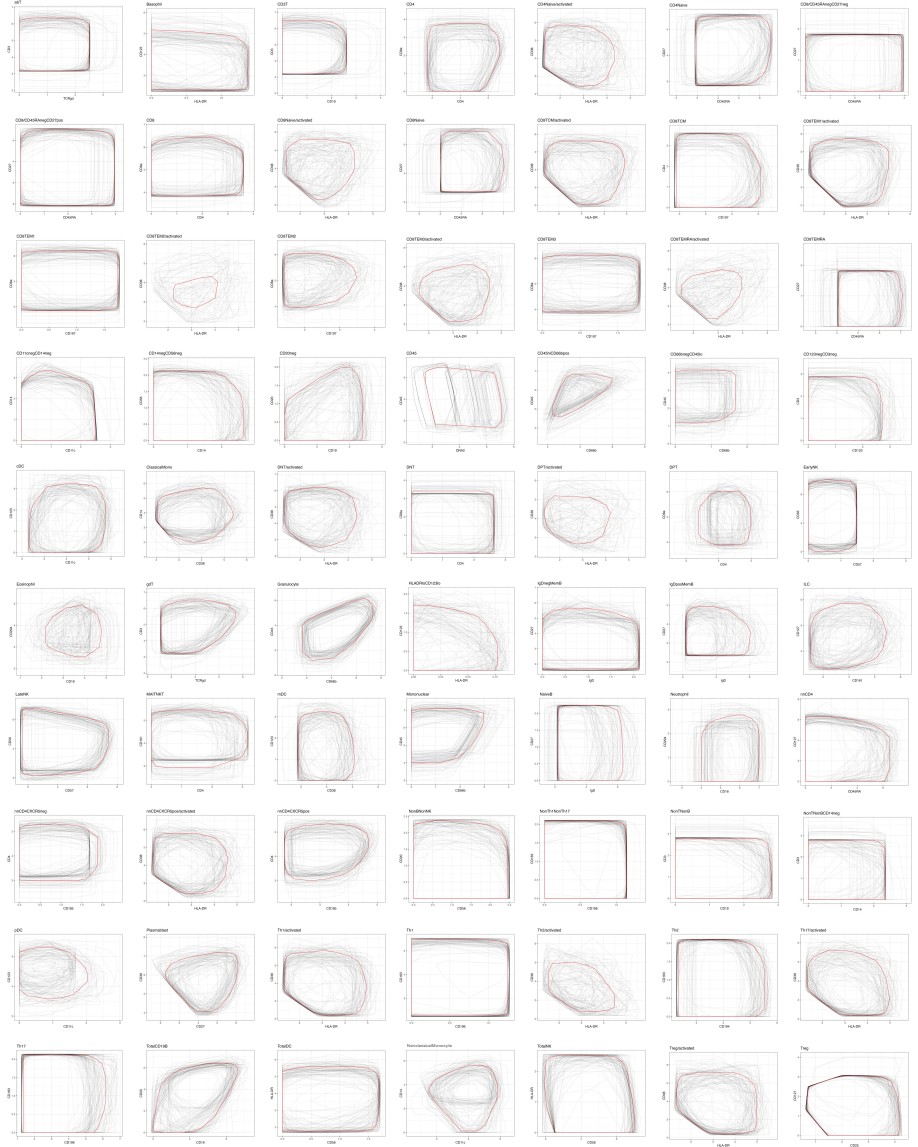

Figure P.3: Consensus gates of statical approach in manual gating.

| | Internal test set | | | | | | | External set 1 | | | | | | | External set 2 | | | | | | |
|---|---|---|---|---|---|---|---|---|---|---|---|---|---|---|---|---|---|---|---|---|---|
| | Ratio | scMAE | GBDT | Static gating | FlowSOM | DNN | CNN | Ratio | scMAE | GBDT | Static gating | FlowSOM | DNN | CNN | Ratio | scMAE | GBDT | Static gating | FlowSOM | DNN | CNN |
| Neutrophil | 6.3e-01 | 0.994 | 0.999 | 0.985 | 0.791 | 0.998 | 0.997 | 6.1e-01 | 0.999 | 0.984 | 0.959 | 0.925 | 1.000 | 1.000 | 8.2e-01 | 0.938 | 1.000 | 0.848 | 0.121 | 0.996 | 0.940 |
| CD66bnegCD45lo | 1.7e-01 | 0.983 | 0.968 | 0.926 | 0.590 | 0.947 | 0.006 | 1.1e-02 | 0.945 | 0.990 | 0.847 | 0.303 | 0.518 | 0.001 | 5.5e-03 | 0.996 | 0.789 | 0.047 | 0.959 | 0.973 | 0.000 |
| CD4Naive | 4.4e-02 | 0.995 | 0.996 | 0.459 | 0.758 | 0.985 | 0.993 | 7.6e-02 | 0.980 | 0.867 | 0.569 | 0.800 | 0.981 | 0.999 | 3.2e-02 | 0.851 | 0.979 | 0.508 | 0.806 | 0.922 | 0.985 |
| ClassicalMono | 1.9e-02 | 0.979 | 0.992 | 0.877 | 0.306 | 0.999 | 0.940 | 3.5e-02 | 0.946 | 0.902 | 0.867 | 0.000 | 0.999 | 0.999 | 2.7e-02 | 0.838 | 0.933 | 0.402 | 0.286 | 0.990 | 0.774 |
| CD8Naive | 1.3e-02 | 0.978 | 0.983 | 0.938 | 0.614 | 0.984 | 0.089 | 3.3e-02 | 0.929 | 0.828 | 0.713 | 0.737 | 0.978 | 0.145 | 1.3e-02 | 0.806 | 0.959 | 0.727 | 0.617 | 0.857 | 0.059 |
| NaiveB | 1.2e-02 | 0.986 | 0.997 | 0.927 | 0.961 | 0.997 | 0.732 | 2.4e-02 | 0.962 | 0.919 | 0.834 | 0.969 | 0.988 | 0.339 | 2.5e-02 | 0.881 | 0.954 | 0.732 | 0.976 | 0.986 | 0.518 |
| Eosinophil | 1.1e-02 | 0.998 | 0.996 | 0.869 | 0.861 | 0.972 | 0.000 | 7.1e-03 | 0.949 | 0.926 | 0.817 | 0.717 | 0.794 | 0.000 | 3.7e-03 | 0.957 | 0.901 | 0.627 | 0.688 | 0.433 | 0.000 |
| EarlyNK | 1.0e-02 | 0.924 | 0.854 | 0.828 | 0.693 | 0.924 | 0.786 | 1.9e-02 | 0.943 | 0.909 | 0.896 | 0.609 | 0.851 | 0.476 | 5.9e-03 | 0.922 | 0.943 | 0.861 | 0.703 | 0.956 | 0.586 |
| LateNK | 1.0e-02 | 0.950 | 0.996 | 0.945 | 0.658 | 0.921 | 0.000 | 1.7e-02 | 0.982 | 0.975 | 0.924 | 0.808 | 0.963 | 0.000 | 4.5e-03 | 0.938 | 0.994 | 0.897 | 0.758 | 0.923 | 0.000 |
| Th1 | 9.2e-03 | 0.963 | 0.949 | 0.933 | 0.721 | 0.902 | 0.000 | 1.8e-02 | 0.989 | 0.903 | 0.948 | 0.745 | 0.940 | 0.000 | 5.6e-03 | 0.891 | 0.994 | 0.900 | 0.681 | 0.947 | 0.000 |
| nnCD4CXCR5pos | 8.4e-03 | 0.979 | 0.986 | 0.977 | 0.536 | 0.920 | 0.026 | 1.7e-02 | 0.955 | 0.862 | 0.948 | 0.642 | 0.916 | 0.009 | 8.1e-03 | 0.850 | 0.981 | 0.838 | 0.566 | 0.868 | 0.079 |
| CD8TEMRA | 7.4e-03 | 0.956 | 0.940 | 0.807 | 0.853 | 0.952 | 0.000 | 1.5e-02 | 0.944 | 0.698 | 0.807 | 0.681 | 0.947 | 0.000 | 4.1e-03 | 0.737 | 0.927 | 0.607 | 0.586 | 0.844 | 0.000 |
| CD45hiCD66bpos | 6.8e-03 | 0.803 | 0.789 | 0.857 | 0.273 | 0.608 | 0.015 | 1.3e-02 | 0.615 | 0.814 | 0.433 | 0.016 | 0.211 | 0.003 | 7.4e-03 | 0.797 | 0.459 | 0.786 | 0.217 | 0.452 | 0.007 |
| Th17 | 6.0e-03 | 0.952 | 0.927 | 0.909 | 0.352 | 0.712 | 0.000 | 1.3e-02 | 0.954 | 0.791 | 0.884 | 0.358 | 0.842 | 0.000 | 6.7e-03 | 0.809 | 0.937 | 0.848 | 0.378 | 0.788 | 0.000 |
| DPT | 4.7e-03 | 0.923 | 0.885 | 0.859 | 0.000 | 0.055 | 0.000 | 1.3e-03 | 0.351 | 0.257 | 0.190 | 0.028 | 0.007 | 0.000 | 7.9e-04 | 0.343 | 0.213 | 0.269 | 0.000 | 0.001 | 0.000 |
| CD8TEM1 | 4.5e-03 | 0.866 | 0.825 | 0.956 | 0.189 | 0.714 | 0.057 | 1.1e-02 | 0.910 | 0.759 | 0.924 | 0.357 | 0.911 | 0.190 | 4.4e-03 | 0.786 | 0.928 | 0.920 | 0.096 | 0.741 | 0.059 |
| gdT | 4.3e-03 | 0.969 | 0.992 | 0.962 | 0.457 | 0.952 | 0.000 | 1.5e-02 | 0.983 | 0.959 | 0.954 | 0.502 | 0.957 | 0.000 | 2.6e-03 | 0.955 | 0.978 | 0.866 | 0.514 | 0.935 | 0.000 |
| CD8TCM | 4.2e-03 | 0.955 | 0.970 | 0.937 | 0.530 | 0.812 | 0.296 | 1.1e-02 | 0.779 | 0.954 | 0.752 | 0.276 | 0.750 | 0.509 | 3.7e-03 | 0.904 | 0.757 | 0.900 | 0.504 | 0.864 | 0.260 |
| Treg | 3.5e-03 | 0.972 | 0.955 | 0.554 | 0.000 | 0.736 | 0.000 | 6.4e-03 | 0.926 | 0.898 | 0.365 | 0.000 | 0.689 | 0.000 | 2.3e-03 | 0.864 | 0.888 | 0.383 | 0.000 | 0.876 | 0.000 |
| Th2 | 3.5e-03 | 0.968 | 0.944 | 0.945 | 0.744 | 0.818 | 0.000 | 5.6e-03 | 0.984 | 0.956 | 0.939 | 0.727 | 0.847 | 0.000 | 3.4e-03 | 0.904 | 0.986 | 0.907 | 0.716 | 0.871 | 0.000 |
| MAITNKT | 2.8e-03 | 0.707 | 0.864 | 0.251 | 0.891 | 0.804 | 0.000 | 7.1e-03 | 0.679 | 0.684 | 0.297 | 0.915 | 0.721 | 0.000 | 1.4e-03 | 0.632 | 0.781 | 0.230 | 0.690 | 0.487 | 0.000 |
| IgDposMemB | 2.3e-03 | 0.982 | 0.987 | 0.809 | 0.000 | 0.352 | 0.000 | 2.7e-03 | 0.532 | 0.431 | 0.404 | 0.000 | 0.280 | 0.000 | 1.6e-03 | 0.429 | 0.484 | 0.352 | 0.000 | 0.149 | 0.000 |
| Basophil | 2.3e-03 | 0.987 | 0.992 | 0.861 | 0.981 | 0.984 | 0.000 | 6.2e-03 | 0.865 | 0.952 | 0.626 | 0.010 | 0.811 | 0.000 | 9.2e-04 | 0.931 | 0.912 | 0.836 | 0.979 | 0.978 | 0.000 |
| IgDnegMemB | 1.9e-03 | 0.947 | 0.980 | 0.901 | 0.966 | 0.892 | 0.000 | 5.3e-03 | 0.737 | 0.855 | 0.678 | 0.727 | 0.674 | 0.000 | 2.9e-03 | 0.832 | 0.728 | 0.809 | 0.899 | 0.759 | 0.000 |
| DNT | 1.5e-03 | 0.882 | 0.879 | 0.447 | 0.135 | 0.507 | 0.000 | 4.5e-03 | 0.926 | 0.851 | 0.599 | 0.031 | 0.782 | 0.000 | 8.8e-04 | 0.826 | 0.942 | 0.456 | 0.046 | 0.508 | 0.000 |
| CD8TEM3 | 1.1e-03 | 0.913 | 0.944 | 0.875 | 0.309 | 0.471 | 0.000 | 4.2e-03 | 0.852 | 0.940 | 0.883 | 0.001 | 0.539 | 0.000 | 2.0e-03 | 0.938 | 0.900 | 0.893 | 0.393 | 0.714 | 0.000 |
| CD4Naive/activated | 8.4e-04 | 0.964 | 0.875 | 0.306 | 0.135 | 0.000 | 0.000 | 7.1e-04 | 0.913 | 0.718 | 0.225 | 0.040 | 0.000 | 0.000 | 9.2e-05 | 0.744 | 0.647 | 0.296 | 0.069 | 0.000 | 0.000 |
| NonclassicalMonocyte | 7.8e-04 | 0.942 | 0.905 | 0.873 | 0.744 | 0.009 | 0.000 | 4.6e-04 | 0.757 | 0.797 | 0.817 | 0.386 | 0.001 | 0.000 | 6.3e-04 | 0.809 | 0.776 | 0.782 | 0.782 | 0.008 | 0.000 |
| CD8TEM1/activated | 6.1e-04 | 0.980 | 0.982 | 0.890 | 0.907 | 0.715 | 0.000 | 1.3e-03 | 0.967 | 0.940 | 0.868 | 0.920 | 0.681 | 0.000 | 9.3e-04 | 0.935 | 0.982 | 0.721 | 0.953 | 0.753 | 0.000 |
| mDC | 4.9e-04 | 0.949 | 0.938 | 0.801 | 0.528 | 0.678 | 0.000 | 1.9e-03 | 0.642 | 0.918 | 0.338 | 0.756 | 0.251 | 0.000 | 1.8e-03 | 0.828 | 0.498 | 0.284 | 0.519 | 0.958 | 0.000 |
| Th1/activated | 4.9e-04 | 0.911 | 0.789 | 0.768 | 0.731 | 0.084 | 0.000 | 4.6e-04 | 0.959 | 0.852 | 0.891 | 0.752 | 0.071 | 0.000 | 5.3e-04 | 0.881 | 0.927 | 0.663 | 0.821 | 0.113 | 0.000 |
| CD8TEMRA/activated | 4.8e-04 | 0.920 | 0.705 | 0.547 | 0.289 | 0.000 | 0.000 | 2.9e-04 | 0.920 | 0.525 | 0.550 | 0.525 | 0.000 | 0.000 | 1.2e-04 | 0.676 | 0.796 | 0.357 | 0.642 | 0.000 | 0.000 |
| Plasmablast | 3.1e-04 | 0.947 | 0.938 | 0.737 | 0.017 | 0.676 | 0.000 | 3.5e-04 | 0.909 | 0.755 | 0.754 | 0.898 | 0.837 | 0.000 | 1.5e-03 | 0.776 | 0.879 | 0.405 | 0.004 | 0.561 | 0.000 |
| Treg/activated | 2.9e-04 | 0.868 | 0.609 | 0.694 | 0.000 | 0.000 | 0.000 | 2.9e-04 | 0.710 | 0.561 | 0.498 | 0.000 | 0.000 | 0.000 | 1.8e-04 | 0.591 | 0.440 | 0.308 | 0.000 | 0.000 | 0.000 |
| CD8Naive/activated | 2.8e-04 | 0.920 | 0.823 | 0.623 | 0.067 | 0.000 | 0.000 | 3.4e-04 | 0.897 | 0.479 | 0.618 | 0.000 | 0.000 | 0.000 | 1.9e-04 | 0.533 | 0.819 | 0.217 | 0.026 | 0.000 | 0.000 |
| CD8TEM2 | 2.4e-04 | 0.823 | 0.819 | 0.675 | 0.053 | 0.034 | 0.000 | 1.2e-03 | 0.890 | 0.879 | 0.702 | 0.519 | 0.057 | 0.000 | 2.7e-04 | 0.920 | 0.893 | 0.725 | 0.078 | 0.036 | 0.000 |
| nnCD4CXCR5pos/activated | 2.4e-04 | 0.937 | 0.802 | 0.640 | 0.147 | 0.000 | 0.000 | 2.3e-04 | 0.858 | 0.686 | 0.722 | 0.098 | 0.000 | 0.000 | 1.4e-04 | 0.733 | 0.883 | 0.496 | 0.038 | 0.000 | 0.000 |
| pDC | 2.1e-04 | 0.974 | 0.986 | 0.831 | 0.980 | 0.962 | 0.000 | 8.1e-04 | 0.130 | 0.946 | 0.108 | 0.838 | 0.157 | 0.000 | 7.3e-04 | 0.866 | 0.139 | 0.590 | 0.964 | 0.766 | 0.000 |
| DNT/activated | 2.0e-04 | 0.883 | 0.528 | 0.677 | 0.179 | 0.003 | 0.000 | 2.5e-04 | 0.909 | 0.323 | 0.746 | 0.235 | 0.000 | 0.000 | 7.1e-05 | 0.567 | 0.653 | 0.415 | 0.129 | 0.000 | 0.000 |
| CD8TCM/activated | 2.0e-04 | 0.975 | 0.902 | 0.754 | 0.000 | 0.000 | 0.000 | 4.9e-04 | 0.498 | 0.915 | 0.375 | 0.000 | 0.000 | 0.000 | 1.3e-04 | 0.956 | 0.471 | 0.599 | 0.000 | 0.000 | 0.000 |
| DPT/activated | 1.9e-04 | 0.918 | 0.655 | 0.521 | 0.715 | 0.000 | 0.000 | 8.4e-05 | 0.258 | 0.169 | 0.150 | 0.115 | 0.000 | 0.000 | 2.2e-05 | 0.417 | 0.146 | 0.212 | 0.041 | 0.000 | 0.000 |
| Th17/activated | 1.8e-04 | 0.943 | 0.618 | 0.701 | 0.618 | 0.001 | 0.000 | 2.2e-04 | 0.954 | 0.662 | 0.755 | 0.759 | 0.001 | 0.000 | 9.8e-05 | 0.842 | 0.716 | 0.606 | 0.751 | 0.000 | 0.000 |
| ILC | 1.0e-04 | 0.926 | 0.910 | 0.787 | 0.217 | 0.002 | 0.000 | 3.0e-04 | 0.520 | 0.819 | 0.334 | 0.392 | 0.001 | 0.000 | 1.4e-04 | 0.817 | 0.452 | 0.325 | 0.228 | 0.000 | 0.000 |
| Th2/activated | 8.0e-05 | 0.952 | 0.542 | 0.454 | 0.002 | 0.000 | 0.000 | 5.1e-05 | 0.970 | 0.718 | 0.588 | 0.076 | 0.000 | 0.000 | 4.6e-05 | 0.920 | 0.698 | 0.431 | 0.000 | 0.000 | 0.000 |
| CD8TEM3/activated | 7.3e-05 | 0.917 | 0.899 | 0.599 | 0.002 | 0.000 | 0.000 | 1.4e-04 | 0.860 | 0.921 | 0.561 | 0.002 | 0.000 | 0.000 | 1.5e-04 | 0.931 | 0.846 | 0.540 | 0.006 | 0.000 | 0.000 |
| CD8TEM2/activated | 8.3e-06 | 0.663 | 0.543 | 0.089 | 0.000 | 0.000 | 0.000 | 5.2e-05 | 0.829 | 0.852 | 0.056 | 0.009 | 0.000 | 0.000 | 8.8e-06 | 0.944 | 0.762 | 0.130 | 0.000 | 0.000 | 0.000 |

Figure P.4: Full results of cell type prediction.

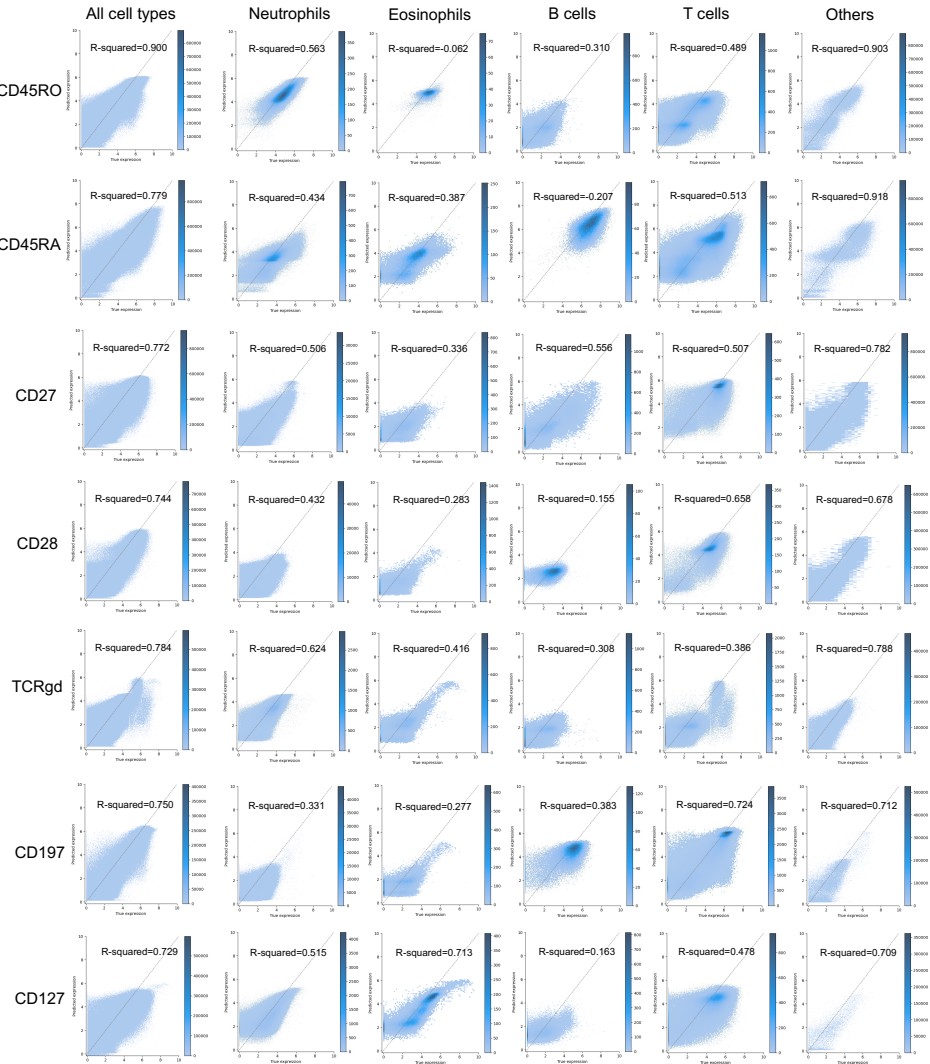

Figure P.5: The true expression and imputed expression plots for the Acute dataset.

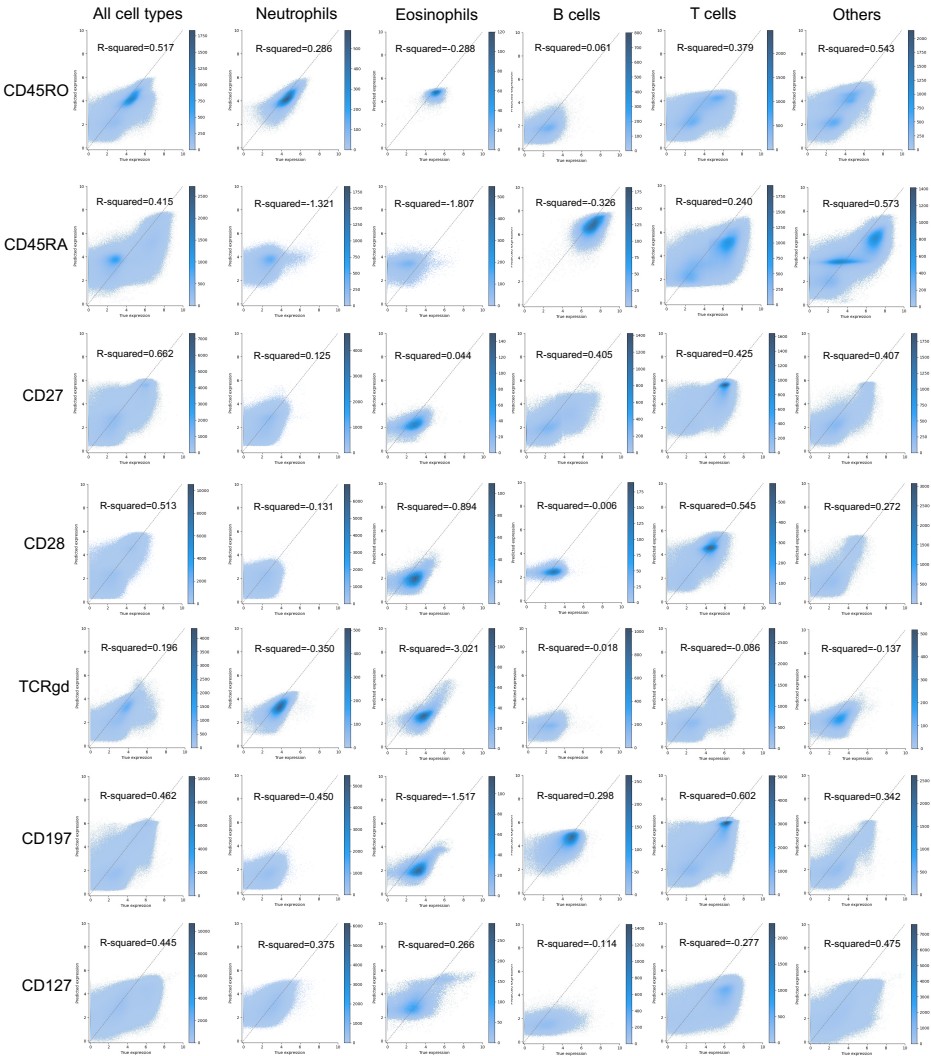

Figure P.6: The true expression and imputed expression plots for the Vaccine dataset.

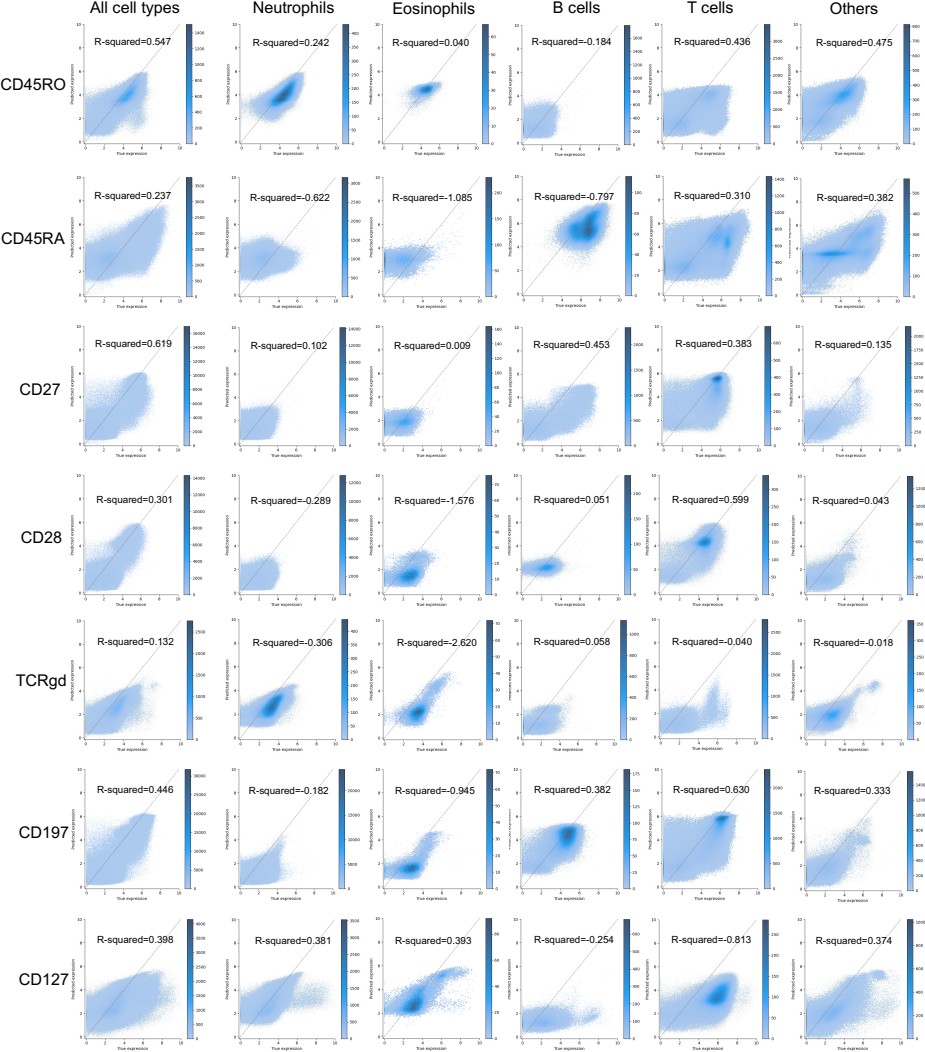

Figure P.7: The true expression and imputed expression plots for the iSPY dataset.

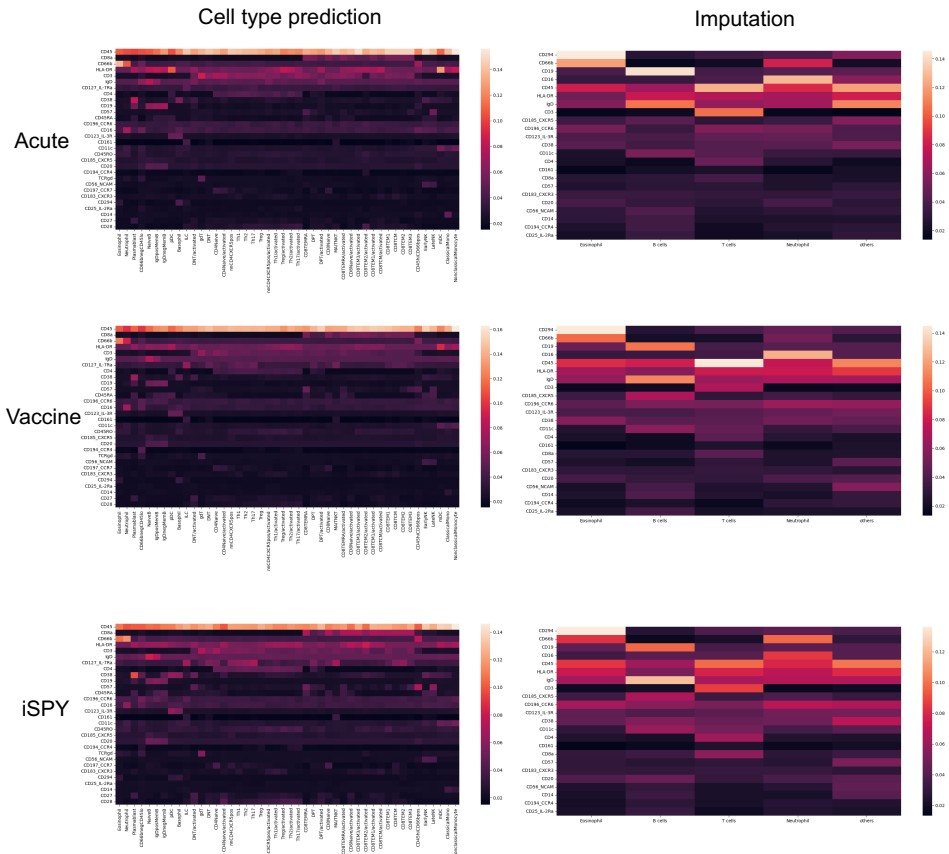

Figure P.8: Attention scores of each cell type in the cell type classification and the imputation task.

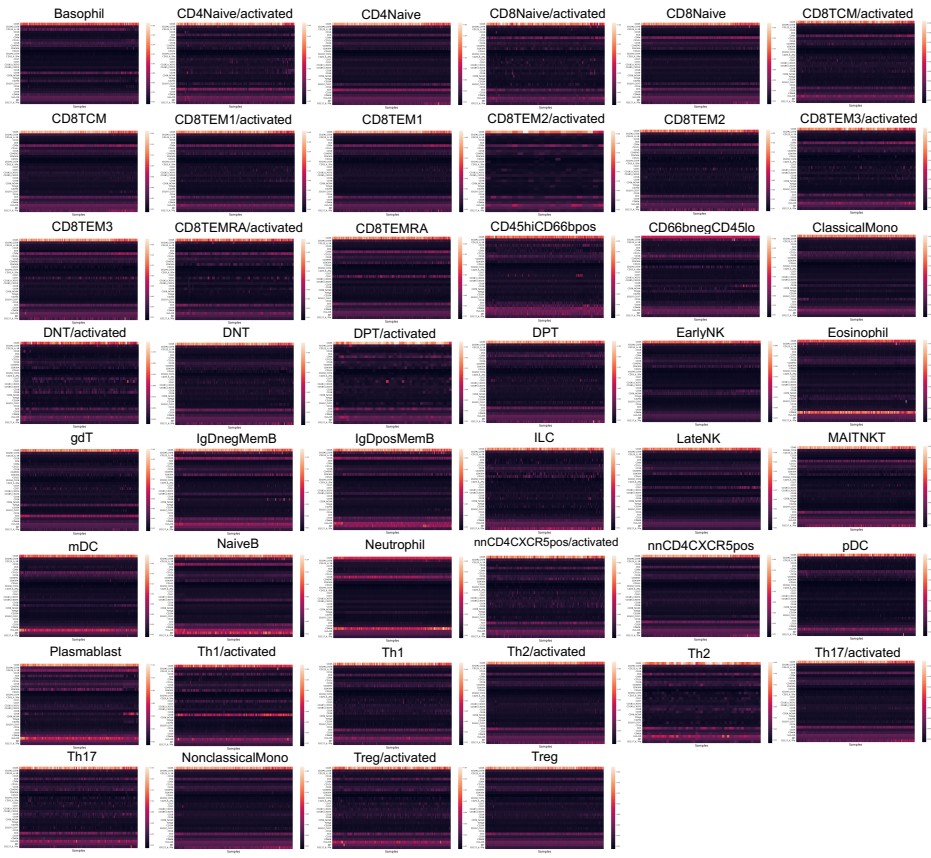

Figure P.9: Attention scores of each cell type for entire samples in the cell type classification task.



Figure P.10: Attention scores of each cell type for entire samples in the imputation task.

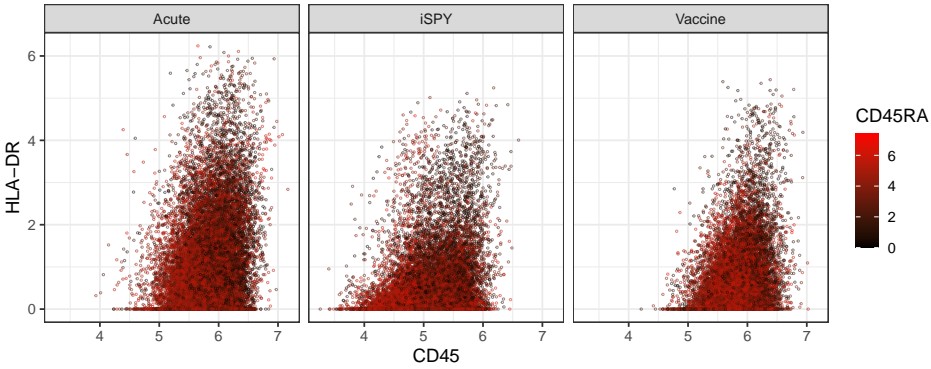

Figure P.11: CD45 and HLA-DR both ßare negatively correlated to CD45RA in T cells.

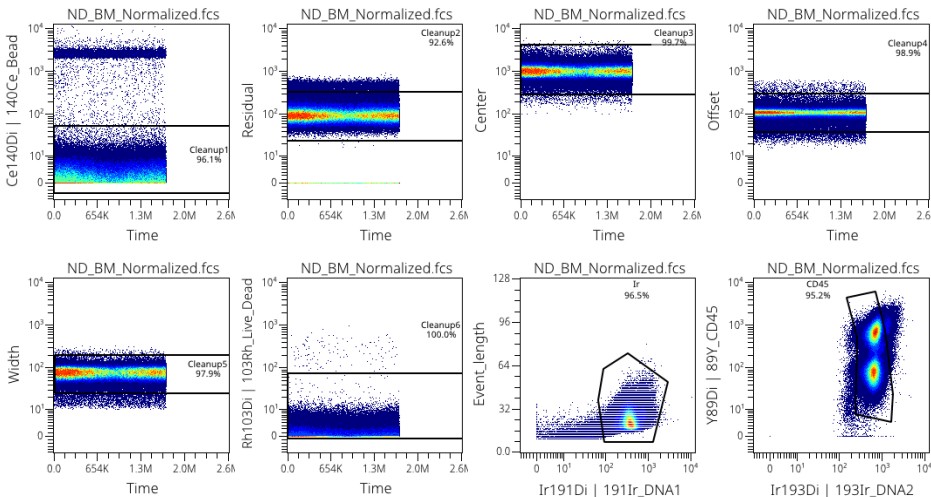

Figure P.12: A standard cleanup procedure, which is a routine manual gating practice. fcs files were gated for beads, debris, doublets, and dead cells using the OMIQ platform.

