# OpenReview forum: "Single-cell Masked Autoencoder: An Accurate and Interpretable Automated Immunophenotyper"
_NeurIPS.cc/2023/Workshop/AI4Science — NeurIPS2023-AI4Science Poster_

### Official Review · Reviewer_gCBs · 2023-10-23
**A good approach in analyzing cytometry data for immune cells**

**Rating:** 6
**Confidence:** 4

**Review:**

The authors proposed scMAE, which adapts the pre-training fine-tuning strategy on high-throughput single-cell cytometry data for immune cells. This model is first pre-trained with a Transformer-based model to adapt informative embeddings for each cell from protein marker expressions. Then the embeddings can be applied to various downstream tasks such as cell type prediction and marker expression imputation.

The method outperforms traditional methods including bivariate plots, unsupervised clustering (e.g., FlowSOM), and CNN/DNN in many aspects, such as higher accuracy in rare cell types, good generalization across experiments, and potential in few-shot learning. The paper is well-written and easy to follow.

Pros:

1 The BERT-like pre-training fine-tuning strategy leverages the large-scale unlabeled data to boost performance, which is innovative in the field of cell cytometry. This method also has potential in higher-dimensional scRNA-seq data as well.

2 The experiment design of cell type prediction is very reasonable. The results that scMAE achieves high performance in cross-dataset evaluations demonstrate the model can handle batch effect and experimental biases.

3 Although attention scores provide limited information and it is doubtful whether this can be regarded as “interpretation”, the learned correspondence between markers shows that the model extracts biologically reasonable relationships.

Cons:

1 Although scMAE achieves the highest performance in Figures 2a&b, it is not clear whether the superior performance comes from the pre-training fine-tuning strategy or the Transformer structure. An additional method that uses Transformer encoder + cell type classifier but only trained with the internal dataset (no pre-training) should also be compared.

2 The cell-type prediction task are evaluated in the situation that all train/test dataset have the same group of profiled marker proteins. In real situations, when we get different groups of marker proteins, how will the model performance be? Since the imputation has a great performance, we may infer the performance still be good, but some systematic evaluations might be helpful.

---

### Official Review · Reviewer_cAJD · 2023-10-24
**A masked autoencoder for automatic gating and imputing mass cytometry data**

**Rating:** 5
**Confidence:** 3

**Review:**

The author built a masked autoencoder, scMAE, for mass cytometry data processing. They demonstrated the performance of scMAE on cell type annotations and imputation tasks, compared to off-the-shelf classifiers (boosted decision trees, deep neural networks), manual gating (rules), and unsupervised methods (FlowSom).

# Quality, clarity
The paper is generally easy to follow, although the description of the model is unclear, confuse, and hard to understand, e.g., the authors used $V_{i,masked} \in \mathbb{R}^{rc\times1}$ to denote the masked protein expressions of cell $i$, while there is no description of $c$ and it's unclear if $rc$ means $r \times c$. A better Fig.1a combined with the clear math descriptions will be helpful.


# Originality and significance
Although transformer-based language models have been developed for single-cell RNA-Seq data, but not for mass-cytometry data. As mass-cytometry data are much cost-effective to generate compared to scRNA-seq data, large language models pre-trained on such data for down-stream analysis (e.g., cell type annotation in the few-shot learning setting) could be useful.


# Pros
* Introduced masked autoencoders for mass cytometry data processing
* The model worked reasonably well in benchmarking studies
* The authors showed that pre-training helped

# Cons
* The masked autoendoder did pretty bad on some distinct cell types (e.g., pDC) while off-the-shelf classifiers work well (e.g. boosting). The authors did not discuss these results.
* Only predictions, no uncertainty quantifications etc, so it's unclear how practitioner can trust the predictions.
* The authored focused on common tasks of cell type annotations and imputation, but did not explore or discuss other utilities of pre-trained model for mass cytometry data processing. The authors can highlight the novelty of their approach.

---

### Meta-Review · Area_Chair_4xrX · 2023-10-27

**Recommendation:** Accept (Poster)
**Confidence:** 4

**Metareview:**

The paper is well-written and interesting, but specific details need improvement and require further clarification. Reviewers have raised concerns about the reported results, which require additional explanation in some areas. Also, there is a need for a more thorough discussion regarding the novelty of their approach and the exploration of other potential applications of pre-trained models in mass cytometry data processing. It would greatly benefit the paper if the authors could also provide more extensive discussions about the model's performance in real-world settings, especially when different groups of marker proteins are involved, as suggested by the reviewers.